# From Biology to Treatment of Monoclonal Gammopathies of Neurological Significance

**DOI:** 10.3390/cancers14061562

**Published:** 2022-03-18

**Authors:** Andrea Visentin, Stefano Pravato, Francesca Castellani, Marta Campagnolo, Francesco Angotzi, Chiara Adele Cavarretta, Alessandro Cellini, Valeria Ruocco, Alessandro Salvalaggio, Alessandra Tedeschi, Livio Trentin, Chiara Briani

**Affiliations:** 1Hematology and Clinical Immunology Unit, Department of Medicine, University of Padova, 35128 Padova, Italy; stefano.pravato@aopd.veneto.it (S.P.); francesco.angotzi@studenti.unipd.it (F.A.); chiaraadele.cavarretta@studenti.unipd.it (C.A.C.); alessandro.cellini@studenti.unipd.it (A.C.); valeria.ruocco@studenti.unipd.it (V.R.); livio.trentin@unipd.it (L.T.); 2Neurology Unit, Department of Neurosciences, University of Padova, 35128 Padova, Italy; francescacastellani2@gmail.com (F.C.); marta.campagnolo@unipd.it (M.C.); alessandro.salvalaggio@aopd.veneto.it (A.S.); chiara.briani@unipd.it (C.B.); 3ASST Grande Ospedale Metropolitano Niguarda, Niguarda Cancer Center, 20162 Milano, Italy; alessandra.tedeschi@ospedaleniguarda.it

**Keywords:** monoclonal gammopathies of neurological significance (MGNS), anti-myelin-associated-glycoprotein (MAG) polyneuropathy, POEMS syndrome, Castleman’s disease, ibrutinib, rituximab, venetoclax

## Abstract

**Simple Summary:**

The peripheral nervous systems may be involved by several hematological diseases ranging from preneoplastic diseases to overt malignancies or paraneoplastic syndromes. In most cases, a monoclonal paraprotein plays a pivotal role in the damage of peripheral nervous systems through different mechanisms. For these reasons, the multidisciplinary approach between hematologist and neurologist is fundamental to correctly diagnose and treat monoclonal gammopathies of neurological significance. We reviewed the biologic, clinic and neurophysiological features, as well as tailored treatments of monoclonal gammopathies of neurological significance.

**Abstract:**

Monoclonal gammopathy and peripheral neuropathy are common diseases of elderly patients, and almost 10% of patients with neuropathy of unknown cause have paraprotein. However, growing evidence suggests that several hematological malignancies synthesize and release monoclonal proteins that damage the peripheral nervous system through different mechanisms. The spectrum of the disease varies from mild to rapidly progressive symptoms, sometimes affecting not only sensory nerve fibers, but also motor and autonomic fibers. Therefore, a multidisciplinary approach, mainly between hematologists and neurologists, is recommended in order to establish the correct diagnosis of monoclonal gammopathy of neurological significance and to tailor therapy based on specific genetic mutations. In this review, we summarize the spectrum of monoclonal gammopathies of neurological significance, their distinctive clinical and neurophysiological phenotypes, the most relevant pathophysiological events and new therapeutic approaches.

## 1. Introduction

Monoclonal gammopathies of neurological significance are a heterogenous group of rare diseases characterized by the presence of a B-cell clone (B lymphocytes and/or plasma cells), a paraprotein (IgM, IgG, IgA or light chains only) and involvement of the peripheral nervous system (PNS). The PNS may be affected during the course of either preneoplastic diseases such as monoclonal gammopathies of undetermined significance (MGUS), or neoplastic hematological diseases such as immunoglobulin light-chain (AL) amyloidosis, POEMS syndrome (Polyneuropathy, Organomegaly, Endocrinopathy, Monoclonal gammopathy, Skin changes), non-Hodgkin lymphomas (NHL) and Castleman’s disease [1].

Several different pathobiological mechanisms sustain either the B-cell or the plasma cell clones. In recent years, next-generation sequencing (NGS) technologies have allowed to shed light on these pathogenetic mechanisms and to identify pivotal somatic mutations that sustain the survival and expansion of pathological B-cells. Accordingly, novel drugs have been developed, which are able to target these key proteins, and have been approved for several hematological diseases. Interestingly, some of these therapies may be used for the treatment of patients with monoclonal gammopathy of neurological significance. 

We review the most relevant pattern of monoclonal gammopathy of neurological significance, with a focus on the known or supposed pathogenic mechanisms that could be targeted by novel therapies.

## 2. Signaling Pathways in B-Cell Malignancies

The B-cell receptor (BCR), the toll-like receptors (TLR), and the C-X-C chemokine receptor type 4 (CXCR4) pathways are key signaling pathways which tune the survival, proliferation, activation, migration and metabolism of B-cell biology.

The BCR is a major actor in the regulation of B-cells biology, as well as in the oncogenesis of B-cell-derived NHL (Figure 1). After antigen binding to the BCR variable regions, the Bruton’s tyrosine kinase (BTK) is activated, promoting the proteasomal degradation of IkB (inhibitor of NF-kB), which subsequently modulates NF-kB activation and downstream proteins [2]. For this reason, ibrutinib, the first in class covalent BTK inhibitor, but also acalabrutinib and zanubrutinib, showed prolonged responses within clinical trials and became the standard of care for most chronic B-cell malignancies. The sustainment of the neoplastic lymphocytes is not only favored by activation of survival and proliferation signaling pathways, but also by disruption of anti-apoptotic proteins. BCL2 (B-cell lymphoma 2) protein is the leading anti-apoptotic protein regulating the BAX-BAK mitochondria channel, the release of cytochrome C and the activation of the intrinsic pathway of apoptosis [3]. BCL2 is overexpressed in several hematological malignancies such as chronic lymphocytic leukemia (CLL) and aggressive NHL, and is targeted by venetoclax [3]. Venetoclax acts binding BCL-2, displacing Bim and other BH-3 only proteins from BLC2. Bim can bind to BAX-BAK, favoring the activation of apotosis [3].

After binding of pathogen-associated molecular patterns (PAMP) to the TLR and recruitment of myeloid differentiation primary response 88 (MYD88) and IRAKs forming the “myddosome complex”, the transcriptional factor NF-kB is activated (Figure 1). Ngo and colleagues identified for the first time somatic activating mutations in MYD88, leading to change of the 265th amino-acid MYD88 (MYD88^L265P^) from leucine to proline [4]. Several other studies confirmed recurrent MYD88^L265P^ mutation in B-cell NHL, by using conventional Sanger sequencing, allele-specific polymerase chain reaction (PCR) analysis, but also NGS technologies on bone marrow cells [4,5]. The MYD88^L265P^ mutation was found in 90% of patients with IgM Waldenstron’s macroglobulinemia (WM)/lymphoplasmocytic lymphoma (LPL), in 55% of non-IgM LPL (55%) and almost half of IgM MGUS [6]. In central nervous system lymphomas, orbital/vitreoretinal lymphomas, intravascular large B-cell lymphomas, the prevalence of MYD88^L265P^ ranges from 44% to 73% [5,6]. Conversely, MYD88^L265P^ is rarely present in splenic marginal zone lymphoma (7.0%), CLL (2.5%), multiple myeloma (1.5%), and is usually absent in IgG or IgA MGUS [5,6]. Other somatic mutations in the MYD88 gene have been identified, but their impact is largely unknown [7]. The MYD88^L265P^ mutation favors the assemble of the complex, boosting the activation downstream signaling [4].

**Figure 1 cancers-14-01562-f001:**
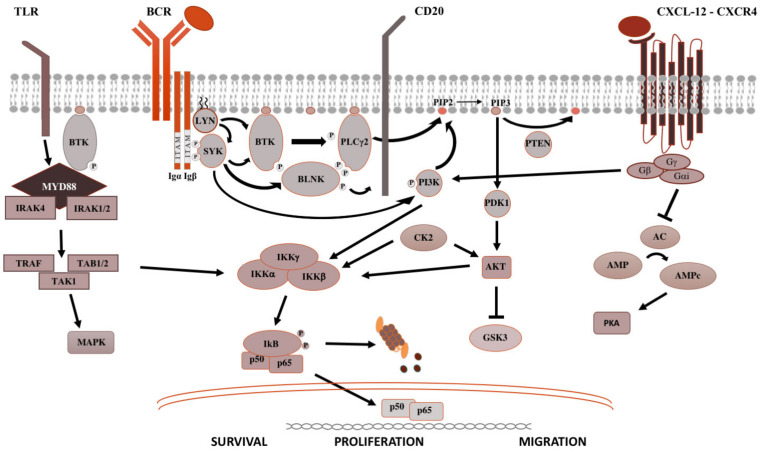
**Pivotal signaling pathways in lymphoproliferative diseases. **On the left side, TLR signaling is shown. Upon recognition of PAMP (pathogen-associated molecular patterns) by TLR (toll-like receptors), MYD88 is recruited to the cell membrane, binds the serine-threonine kinase IRAK proteins (IL-1R associated kinase), TRAF (tumor necrosis factor receptor-associated factor) and TAK1 (transforming growth factor beta-activated kinase 1) [8], which activate the MAPK (mitogen-activated protein kinase) cascade and IKK (inhibitor of the NF-κB kinase). IKK complex is made up of IKKα and IKKβ, with kinasic activity, and the regulatory IKKγ subunit. Once activated, IKK phosphorylates IκB (inhibitor of NF-κB), leading to its ubiquitylation and proteasomal degradation, thereby, the release of the NF-κB (RELA p65–p50 in the classical pathway and RELB-p52 in the alternative pathway). Then, NF-κB subunits translocate into the nucleus and modulate the proliferation and survival of B lymphocytes [2]. In the middle is shown the BCR signaling. After antigen binding to the B-cell receptor (BCR), the Src kinase LYN is activated, phosphorylating CD79A/B and recruiting SYK (spleen tyrosine kinase) [9]. LYN and SYK activate by phosphorylation BTK [10], and, subsequently, the phospholipase PLCγ2, activating NF-kB pathway, MAPK and PI3K pathways. On the right is shown the CXCR4 signaling. After binding of CXCL12 to the seven domains trans-membrane CXCR4 receptor coupled with trimeric G protein, the Gα subunit dissociates from the Gβ/Gγ dimer and inhibits adenyl cyclase (AC), causing the reduction of intracellular cAMP (cyclic AMP) levels and switching off PKA (protein kinase A). Meanwhile, the Gβ/Gγ dimer actives PI3K, modulating lymphocytes migration, and phospholipase PLC, which acts on PIP2 (phosphatidylinositol 4,5-bisphosphate) releasing IP3 (inositol 1,4,5-trisphosphate), favoring the mobilization of Ca2+ from intracellular stores, and diacylglycerol, promoting the activation of PKC (protein kinase C) and MAPK cascade [2].

CXCR4 is one of the most important receptors involved in the regulation of homing and egress of hematological cells trafficking from bone marrow blood and secondary lymphoid organs (Figure 1) [11]. Acquired mutations of CXCR4 gene, such as R334X, G336X, S338X (the most common), E343X, have been described in up to 40%, causing the truncation of the C-terminal tail of this transmembrane protein [12].

Both MYD88 and CXCR4 mutations should be evaluated in bone marrow cells, according to hematological guidelines [12]. However, some findings suggest that MYD88^L265P^ mutation may be evaluated in the peripheral blood cell or by using cell free DNA, thus avoiding invasive diagnostic procedures [13,14]. On the other hand, bone marrow biopsy is recommended in order the correctly diagnose patients’ IgM gammopathy [12].

## 3. Monoclonal Gammopathies of Neurological Significance

The association between peripheral neuropathy and monoclonal protein has been extensively investigated in the literature, being first reported in cases of myeloma and WM [15]. However, the exact prevalence of monoclonal gammopathy of neurological significance is unknown and likely underrated. Monoclonal proteins have been found to be present in 10% of otherwise unexplained peripheral neuropathies [16], with a frequency 6 to 10 times greater than in the general population. The occurrence of both paraprotein and neuropathy increases with age [17] and the association might be casual. Whereas, the most common paraprotein is the IgG isotype, neuropathy, usually demyelinating, is much more common in patients carrying an IgM paraprotein (50–75%, compared with 17–35% with IgG and 8–15% with IgA) [15]. When an association between the hematological diseases and the polyneuropathy is suspected, due to the high prevalence of MGUS in the general population, the first step is to exclude common causes of secondary polyneuropathy such as diabetes mellitus, vitamin deficiency, chronic alcohol consumption, drugs, HCV infection, etc. For these reasons, a multidisciplinary assessment involving hematologists, neurologists, radiologists and neurophysiologists should be guided based on the type of the paraprotein IgM vs. IgG/IgA. In selected cases, nerve biopsy (including light microscopic examination, immunocytochemical techniques and electron microscopic imaging) may be necessary, especially when suspecting vasculitis or neurolymphomatosis (see below). Summary flow charts for IgM and IgG/IgA monoclonal-gammopathy-related peripheral neuropathies are shown in Figure 1 and Figure 2.

## 4. Monoclonal Gammopathies of Undetermined Significance

MGUS is a preneoplastic condition which can be found in approximately 3% of people ≥50 years [18], diagnosed by the presence of (i) monoclonal protein <30 g/L, (ii) <10% clonal neoplastic cells within the bone marrow, (iii) lack of organomegaly and end-organ damage, according to the Mayo clinic guidelines. MGUS can be caused by the presence of a full antibody (heavy-chain and light-chain), further classified according to the type of the heavy chain in IgG, IgM, IgA and IgD MGUS, and/or light-chain-only. 

It can progress to overt neoplasia and is associated with shorter survival than age-matched control population [18]. While almost all patients with multiple myeloma and WM usually had an MGUS phase, not all people with MGUS will develop a hematological disease. The most common MGUS subtype is IgG MGUS, but it is associated with a low risk of developing multiple myeloma—1% people/year if monoclonal protein is <15 g/L and the free–light chain ratio is within normal range [18]. Conversely, IgM MGUS harbors the highest risk of evolution to indolent lymphoma such as WM, marginal zone lymphoma or CLL (described afterwards) [18]. 

The knowledge of the genomic profiling of IgM gammopathies has favored the achievement of significant therapeutic improvement. Gain-of-function L265P mutation of MYD88, the most commonly mutated gene in WM and IgM-MGUS [5], favors the activation of BTK and of NF-κB signaling (Figure 1). The CXCR4 gene is less commonly mutated, usually by nonsense or frameshift mutations, but it has been shown to be associated with lower response rates and early relapse [12] (Figure 1). Recently, Varettoni et al. [19] found that MYD88-mutated patients with at least 10 g/L of paraprotein have the highest risk of progression to WM or other lymphoproliferative disease (11.6% at 5 years and 38% at 10 years from the IgM MGUS diagnosis).

In almost half of the cases, the IgM reacts against myelin-associated glycoprotein (MAG) [20,21]. Anti-MAG neuropathy is described below. IgM paraproteinemic neuropathy may present with antibodies to peripheral nerve antigens different from MAG [22], or it may not present antibody reactivity at all [23,24]. In such cases, also with a rapid neurological progression, pain, early prominent motor involvement or axonal damage, AL amyloidosis or cryoglobulinemic vasculitis should be considered. Sometimes, in patients with a multifocal motor neuropathy with conduction block phenotype, an IgM paraprotein with antiganglioside GM1 antibodies can also be detected [15]. Bing–Neel syndrome is a rare complication of WM, caused by the infiltration of neoplastic B-cells in the cerebral parenchyma, proximal nerve roots, peripheral nerves and meninges. Recurrent manifestations include headaches, cognitive and psychiatric dysfunction, seizures, cranial and peripheral neuropathies. MYD88^L265P^ can be found in the cerebral spinal fluid of almost all patients [25]. Despite this aggressive presentation, Bing–Neel syndrome responds well to BTK inhibitors [26].

## 5. Waldenström’s Macroglobulinemia/Lymphoplasmacytic Lymphoma

Waldenström’s macroglobulinemia is characterized by the proliferation of mature lymphoplasmacytic B-cells which release macroglobulin, most commonly an IgM paraprotein or more rarely an IgA or IgG subtype. While MYD88^L265P^ mutation is found in almost all patients with IgM-WM, CXCR4^WHIM-like^ mutation is found in less than one-third of patients, usually at subclonal level, and is associated with a larger disease burden, lower response to ibrutinib, and usually decreased overall survival due to early relapse [5,12].

Peripheral axonal and/or demyelinating neuropathies are common in WM, being found up to 62.5% of patients, with different manifestations depending on the underlying pathogenic mechanism. Demyelinating polyneuropathy in WM is commonly symmetric, distal, slowly progressive and associated with IgM antibodies against nerve antigens [22,27], most commonly MAG. The presence of polyneuropathy is a criterion for WM treatment. Patients with MW may also develop axonal polyneuropathies caused by drugs such as bortezomib, cryoglobulins, light-chain amyloid deposition, B-cell infiltration, and rare CANOMAD disease characterized by chronic ataxic neuropathy, ophthalmoplegia, IgM M-protein, positivity for cold agglutinins and antibodies to disialosyl antigens (GD1b, GQ1b, GT1a, GT1b) [28].

Cryoglobulinemias can be classified as follows: type I—monoclonal immunoglobulin (more common IgM > IgG > IgA); type II—monoclonal IgM and polyclonal IgG; type III—polyclonal IgM with rheumatoid factor activity and polyclonal IgG. While type II is more common in WM, type III is seen predominantly in connective tissue diseases or in chronic infections such as chronic HCV hepatitis [29]. Monoclonal IgMs cryoglobulins may precipitate below 37 °C, causing severe painful multifocal neuropathy, sometimes involving cranial nerves. Patients can also suffer from arthralgia, glomerulonephritis, skin ulceration or purpura.

## 6. Anti-Myelin-Associated Glycoprotein (MAG) Antibody Neuropathy

Polyneuropathy with IgM anti-MAG antibodies and IgM MGUS, but also WM, marginal zone lymphoma (MZL) or CLL [1] is the most common IgM paraproteinemic polyneuropathy [30]. Anti-MAG antibodies have a pathogenic role, as demonstrated by experimental mice models and human sural nerve biopsies, showing IgM and complement deposition, as well as characteristic widening of myelin lamellae on electron microscopy investigation leading to the segmentation of myelin [1].

Patients usually present with a length-dependent slowly progressive symmetric polyneuropathy, with predominant sensory impairment at lower limbs, resulting in sensory ataxia, and tremor at upper limbs, with a significant impact on patients’ disability and quality of life [31]. Motor involvement generally occurs late in the course of the disease. Regarding neurophysiological features, a demyelinating neuropathy with symmetric reduction of conduction velocities and sensory nerve action potentials is observed with the typical finding of disproportionately prolonged distal motor latencies and lack of conduction blocks. Axonal features may appear in addition to demyelination in patients with a long disease duration. 

To date, inadequate evidence has been obtained from therapeutic trials [32]. After initial results from small studies, rituximab, an anti-CD20 chimeric monoclonal antibody, was evaluated in two randomized clinical trials with controversial results [33,34]. Remarkably, patients with WM were excluded from these trials. Real-life retrospective studies agree that rituximab is active in almost 30–50% of the patients [33,34], with the same rate of responses in patients with WM or IgM MGUS [35]. Albeit with low-quality evidence, a Cochrane meta-analysis [32] confirmed that rituximab improves disability scales and the response to questionnaires in the global impression of anti-MAG antibody neuropathy. Therefore, rituximab is currently used in the clinical practice either alone or in combination with cyclophosphamide [36], fludarabine [37] or bendamustine [38]. Obinutuzumab, a humanized glycoengineered anti-CD20 monoclonal antibody, has also been used as a possible alternative treatment in patients with anti-MAG antibody neuropathy, with controversial results and concerns regarding possible toxicity [39,40].

Recently, ibrutinib has proven to be active in 3 MYD88^L265P^-mutated and CXCR4 wild-type patients with anti-MAG antibody polyneuropathy and WM. Two of the three cases were also refractory to rituximab [41]. In particular, decrease of INCAT (Inflammatory Neuropathy Cause and Treatment) Disability and INCAT Sensory Sum (ISS) scores confirmed a neurological improvement. After 12 months of treatment, ibrutinib was safe in these elderly patients, and no events of atrial fibrillation or infections were recorded. In addition, second-generation BTK inhibitors, such as acalabrutinib [42] and zanubrutinib [43], showed promising results as single agents or in combination with rituximab for the treatment of symptomatic WM. These new drugs selectively target BTK and are associated with lower adverse events than ibrutinib [44]. Zanubrutinib displayed also encouraging results in MYD88 wild-type patients [45]. Venetoclax is an oral and selective BCL2 inhibitor that in combination with rituximab proved to be highly active in B-cell malignancies even after ibrutinib failure [46]. Recently, Castillo et al. showed that venetoclax was able to induce remission in relapsed WM regardless of CXCR4 mutations [47].

## 7. Immunoglobulin Light-Chain (AL) Amyloidosis

AL amyloidosis is caused by the misfolding and deposition of immunoglobulin light chains in tissues and organs released by lymphoplasmacytic cells in patients with MGUS, multiple myeloma or, less commonly, WM or CLL. Patients with AL amyloid neuropathy may display neuropathic pain and autonomic symptoms with orthostatic hypotension, diarrhea or constipation, pupil asymmetry, genitourinary and sexual dysfunction. In the diagnostic workup, it is useful to look for damages to heart (heart failure with septum thickness and preserved ejection fraction), kidney (albuminuria) and intrahepatic cholestasis (increase of bilirubin and alkaline phosphatase). Additional red flags in the diagnosis of AL amyloidosis are coagulation factor X deficiency, highly specific to AL amyloidosis but uncommon, sometimes favoring the development of “racoon eyes” (i.e., periorbital ecchymoses) [48], and macroglossia. All these points make AL amyloidosis diagnostic delay quite common [48]. 

Isolated peripheral nervous system involvement is rare, therefore, AL amyloidosis should be suspected in patients with axonal polyneuropathy, history of carpal tunnel syndrome, pain and autonomic dysfunctions, and instrumental or biochemical markers suggestive of heart, kidney and/or liver damage. Notably, the monoclonal component in amyloidosis generally carries the lambda light chain, except for the IgM-related amyloidosis [49]. Nerve biopsy with Congo red staining and electronic microscopy are the gold standard in demonstrating the amyloid deposits in and around endoneurial blood vessel walls [50]. Nerve biopsy helps also to differentiate amyloid neuropathy from the rare heavy-chain IgM deposition neuropathy, which is not characterized by autonomic symptoms [15]. The pathological process starts with axonal degeneration of unmyelinated and small myelinated fibers, but in the later phase of the disease, fibers of all sizes become involved [51,52]. Moreover, increased insight into the biology of AL amyloidosis helped identify target treatments. For example, thalidomide, lenalidomide and pomalidomide have immunomodulatory effects on the bone marrow niche though inhibition of cereblon, a key protein of the E3 ubiquitin complex, leading to ubiquitylation and proteasomal degradation of IkB, ikaros, aiolos and casein kinase 1 [2]. Bortezomib and carfilzomib inhibit the proteasome causing the accumulation of IkBα, the inhibition of NF-kB transcriptional factor, apoptosis of plasma cells, and decrease of IL-6 and Vascular Endothelial Growth Factor (VEGF) by microenvironment [53]. Daratumumab, an anti-CD38 monoclonal antibody, has become the backbone of treatment of multiple myeloma and AL amyloidosis, being able to quickly drop the free light chains [54]. It is unknown whether immunomodulatory drugs might have a direct amyloid targeted effect or rather an indirect action on plasma cells together with proteasome inhibitors and monoclonal antibodies. On the other hand, thalidomide and bortezomib could be a concern, because of their known neurotoxicity, in patients with amyloid neuropathy.

## 8. POEMS Syndrome

Crow–Fukase syndrome, also known as POEMS syndrome (Polyneuropathy, Organomegaly, Endocrinopathy, Monoclonal gammopathy, Skin changes), is a rare paraneoplastic disease, characterized by polyneuropathy and monoclonal gammopathy, usually lambda-chain-restricted. Several organs and systems might be involved, causing pulmonary arterial hypertension, edema of the optic nerve, peripheral edema, pleural and pericardial effusions, ascites, white nails, thrombocytosis, polyglobulia, arterial thrombosis, osteosclerotic lesions [55].

Since peripheral neuropathy is often the presenting clinical feature, the disease is frequently misdiagnosed as chronic inflammatory polyradiculoneuropathy (CIDP), but neurophysiological studies display early axonal or mixed damage [56]. The POEMS neuropathy is more commonly symmetric, with subacute or acute onset, often painful, involving both sensory and motor nerve fibers, and rapidly progressing with distal weakness [56].

The pathogenesis is unknown, but VEGF likely plays a pivotal role. In fact, serum VEGF levels correlate with disease activity, nerve damage and hematological findings [57,58,59]. In vitro studies have shown that VEGF, by acting on endothelial cells, increases vascular permeability and promotes angiogenesis. In addition, nerve biopsies confirmed that VEGF is highly expressed in blood vessels [59], further supporting the pathogenetic role of VEGF in POEMS syndrome.

Recently, Nagao et al. found recurrent mutations affecting KLHL6, LTB, EHD1, EML4, HEPHL2, HIPK2, and PCDH10 by whole-exome sequencing in patients with POEMS syndrome [60]. Conversely, mutations of RAS, NF-kB and MYC pathways, usually found in multiple myeloma, were not found in POEMS [61]. These findings support different pathogenetic mechanisms in POEMS-associated gammopathy and multiple myeloma.

Patients with POEMS syndrome often show subclinical pachymeningeal involvement [62,63]. Histology shows meningothelial cells proliferation, endothelial hyperplasia, thickening of the media of the arterioles, sometimes causing also neovascularization, likely mediated by increase VEGF levels [62]. No inflammatory signs are present, therefore, the term pachymeningitis should not be used [62]. The improvement of pachymeningeal involvement with lenalidomide further confirm the key role of VEGF on meningeal neoangiogenesis [64]. Neovascularization in the skin biopsy and in the perineurium have also been observed in patients with POEMS syndrome [65].

Despite the strong relationship between disease activity and VEGF levels, anti-VEGF antibodies showed disappointing results in POEMS syndrome [55,66]. Radiotherapy is active only in patients with focal bone lesion and without multiple myeloma [55]. Instead, patients with disseminated bone marrow involvement are treated with protocols derived from the treatment of multiple myeloma or AL amyloidosis [55,66]. Young patients without significant comorbidities should receive autologous stem cell transplantation conditioned with high-dose melphalan, which is able to achieve hematological remissions in almost 80% of patients [67], neurophysiological and objective neurological improvement in almost all patients [66,68,69,70]. Conversely, transplant-ineligible patients or relapsed cases should receive immunomodulating agents or proteasome-based treatments. Lenalidomide led to good results both in treatment-naive and relapsed patients [64,71,72], however, almost all patients relapse after almost 6 months following the end of therapy [66]. Bortezomib, the first in the class of proteasome inhibitors, has been used in almost fifty cases in the literature [66,73,74], and it has been shown to be able to decrease VEGF, paraprotein levels; despite its painful neurotoxicity, it was able improve also the neuropathy. Patients with POEMS syndrome are likely to benefit from a daratumumab-based or second-generation proteasome inhibitor (carfilzomib or ixazomib) treatment considering the high rate of complete remissions and undetectable minimal residual disease, as in patients with multiple myeloma. To date, three patients have been treated with a daratumumab-based combination with decrease of VEGF, hematological and neurological improvement [55,75]. One patient has also been treated successfully with anti-BCMA chimeric antigen receptor (CAR)-T cells [76].

## 9. Castleman’s Disease

Angiofollicular lymph node hyperplasia, also known as Castleman’s disease, is a rare nonmalignant lymphoproliferative disease. Castleman’s disease manifestations can range from unicentric disease, usually showing as a slowly increasing mass mainly in the lung, to multicentric subtype with lymphadenopathy and systemic symptoms, to the TAFRO syndrome (thrombocytopenia, anasarca, fever, reticulin fibrosis in the bone marrow and organomegaly), the latter being associated with a worse prognosis [77]. Fifty percent of cases are idiopathic, while the remaining occur during HIV and/or human herpes simplex virus 8 (HHV8) infection [78]. Castleman’s disease can be isolated or associated with POEMS syndrome. 

Polyneuropathy, mainly demyelinating and sensitive, can be found in up to 30% of patients with Castleman’s disease, regardless of the coexistence of POEMS syndrome. It is usually less severe than the neuropathy associated with POEMS syndrome [79]. In Castleman’s disease patients, IL-6 is the predominant cytokine and is a marker of disease activity [77].

Several treatments have been used in Castleman’s diseases from surgical excision of unicentric diseases to polychemotherapy. Most of the published data derive from case reports and single-center analyses [77]. Siltuximab is the anti-IL-6 monoclonal antibody, approved for the treatment of idiopathic Castleman’s diseases. In the Phase 2 randomized clinical trial, continuous siltuximab treatment showed durable tumor and systemic responses compared to placebo (35% vs. 0%) [80]. Tocilizumab is a monoclonal antibody targeting the receptor of IL-6, able to decrease lymph nodes size and to mitigate systemic symptoms in 86% of cases, even after 5 years [81]. Tocilizumab is approved in Japan for multicentric Castleman’s disease.

## 10. Non-Hodgkin Lymphoma-Related Peripheral Neuropathies

### 10.1. Chronic Lymphocytic Leukemia

CLL is a heterogeneous disease, which mainly affects elderly patients with comorbidities, ranging from indolent forms, which will never need treatment, to symptomatic progressing diseases with a dismal prognosis [82,83,84]. Indolent cases usually harbor a mutated IGHV gene, but not TP53 disruptions, while aggressive cases harbor unmutated IGHV gene, TP53 gene deletion or mutation, and/or complex karyotypes [82,85,86,87]. The key role of CD20 and the BCR in the pathogenesis of CLL is supported by the efficacy of obinutuzumab and BTK inhibitors in the disease [88,89]. Monoclonal gammopathy can be found in almost 15% of CLL, with IgM paraprotein being associated with TP53 deletion or mutation and shorter time to first treatment [90]. Peripheral neuropathy in CLL may be due to different causes, including varicella-zoster virus and HCV infections, iatrogenic, immune-mediated, rarely, neuropathy is secondary to neurolymphomatosis [91,92] or AL amyloidosis [93]. In a single-center retrospective study, we identified 19 (2.2%) patients out of 860 CLL who developed peripheral polyneuropathy. The estimated incidence of new-onset neuropathy was 2.1% and 6.9% after 10 and 20 years from the CLL diagnosis [91]. Fifteen out of the nineteen patients had a sensory or sensorimotor axonal neuropathy, one had a multiple mononeuropathy, and three fulfilled the clinical and neurophysiological criteria of chronic inflammatory polyradiculoneuropathy (CIDP) [94]. Development of peripheral neuropathy was more common in previously treated subjects, with TP53 deletion and serum monoclonal gammopathy [95]. Two CLL patients with concurrent anti-MAG antibody polyneuropathy were treated with six cycles of chlorambucil–obinutuzumab [40], showing persistent hematological and neurological remissions (Table 1). However, both developed pneumonia that required hospitalization [40]. Despite the efficacy, caution is warranted for potential adverse effects in aged patients.

### 10.2. Intravascular Lymphoma

Angiotropic lymphoma or malignant angioendotheliomatosis, also known as intravascular large cell lymphoma (IVL) [96], is a rare but highly aggressive NHL, characterized by infiltration and occlusion of small vessels by neoplastic lymphocytes [97] causing ischemic or hemorrhagic lesions [98]. NGS studies identified MYD88^L265P^ and CD79b mutations in 44% and 26% of IVL, respectively [99]. These mutations made the NF-κB pathway constitutively activated. IVL typically shows central nervous system symptoms/signs such as seizures, myoclonus or meningoradiculitis. PNS involvement is not common in IVL, but neuropathies with sensory and motor deficits may be present, ranging from acute polyradiculoneuropathy [100] and multiple mononeuropathies [101] to demyelinating polyneuropathy [102]. The prognosis of the IVL is poor, due to a high rate of relapse [97]. BTK inhibitors might be a therapeutic option, due to their ability to pass the blood–brain barrier and to be active in central nervous system lymphoma [103]. However, the efficacy of both first- and second-generation BTK inhibitors in IVL is unknown.

### 10.3. Neurolymphomatosis

Neurolymphomatosis is defined as the infiltration of the perineurium, epineurium or endoneurium by lymphomatous cells confirmed by nerve biopsy [104,105,106]. It can be primary if the NHL involves only the PNS, or, more commonly, secondary due to the infiltration of PNS from other organs affected by the lymphoma. Neurolymphomatosis may be the first manifestation of NHL or it may occur at relapse. Neurolymphomatosis may be caused by all NHL, more commonly, aggressive B-cell NHL such as diffuse large B-cell lymphoma (DLBCL), but also CLL [92] or natural killer cell-derived lymphoma [107]. The diagnosis of neurolymphomatosis is commonly missed because of the rarity of the disease, the heterogeneous manifestations, the technical difficulties in establishing a definitive diagnosis and the multifocal involvement of nerves. The most frequent misdiagnosis is CIDP [108] due to the presence of demyelinating signs at neurophysiological studies. Neurolymphomatosis should therefore be considered in patients with unexplained PNS manifestations and peripheral neuropathy who do not fulfill the diagnosis criteria for monoclonal gammopathy of neurological significance or who do not response adequately to treatment. Although neuroimaging is gaining importance in the diagnostic workup of neurolymphomatosis [109,110,111], nerve biopsy remains mandatory for the diagnosis.

## 11. Conclusions

Monoclonal gammopathies of neurological significance include a widespread range of manifestations, ranging from slowly progressive sensitive demyelinating polyneuropathy with anti-MAG antibody to subacute rapidly progressive forms as in POEMS syndrome or neurolymphomatosis. 

The monoclonal gammopathy isotype, the accompanying light chain, the features of the polyneuropathy, the neurophysiological study and sometimes the nerve biopsy findings, as well as the presence of systemic symptoms or biochemical markers, help guide the correct diagnosis (Figure 2 and Figure 3).

A multidisciplinary approach, in particular, involving hematologists and neurologists, is recommended in order to establish the correct diagnosis of monoclonal gammopathy of neurological significance and to tailor therapy also based on recurrent genetic abnormalities.

Although the rarity of some neuropathies makes it difficult to plan randomized clinical trials, increasing evidence supports the efficacy of targeted therapies also in monoclonal gammopathies of neurological significance (Table 1).

## Figures and Tables

**Figure 2 cancers-14-01562-f002:**
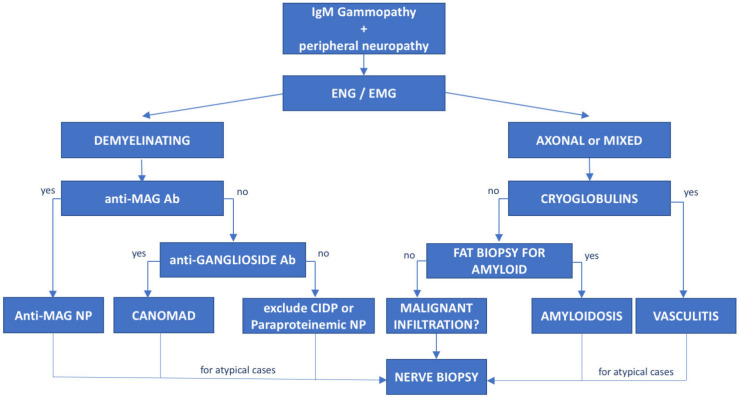
**Diagnostic flowchart of peripheral neuropathy associated with IgM monoclonal gammopathy.** Patients with IgM monoclonal protein and peripheral neuropathy diagnosed with neurophysiological tests (electroneurography (ENG) and electromyography (EMG)) should undergo further investigation based on the type of neuropathy. Demyelinating cases with prolonged distal latency must be tested for anti-MAG antibodies to rule out anti-MAG neuropathy; negative cases with chronic ataxia and ophthalmoplegia must be tested for antiganglioside antibodies to exclude CANOMAD diseases. In double-negative patients, CIDP and paraproteinemic neuropathy should be taken into consideration. On the other hand, for patients with axonal or mixed (axonal and demyelinating features) neuropathies, cryoglobulins and light-chain amyloid deposition in fat biopsy should be investigated. However, for atypical or rapidly progressive cases, nerve biopsy is recommended to exclude neurolymphomatosis. Ab—antibodies, NP—neuropathy.

**Figure 3 cancers-14-01562-f003:**
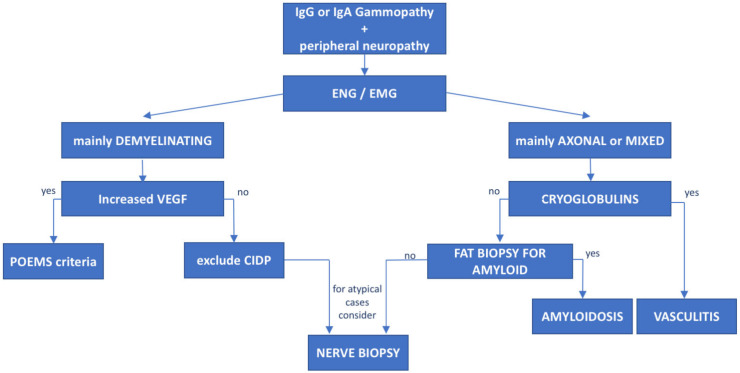
**Diagnostic flowchart of peripheral neuropathy associated with IgG or IgA monoclonal gammopathies.** Patients with IgG or IgA monoclonal protein and peripheral neuropathy diagnosed with neurophysiological tests (electroneurography (ENG) and electromyography (EMG)) should undergo further investigation based on the type of neuropathy. Mainly demyelinating cases with subacute onset should be tested for VEGF levels to rule out POEMS syndrome. In negative cases, CIDP should be taken into consideration. On the other hand, patients with mainly axonal or mixed (axonal and demyelinating) features cryoglobulins and light-chain amyloid deposition in fat biopsy should be assessed. In atypical or rapidly progressive cases, nerve biopsy may be considered.

**Table 1 cancers-14-01562-t001:** Summary of diseases and treatments.

DISEASES		TREATMENTS
**A** **nti-MAG antibody neuropathy**	IgM MGUS	Rituximab
	CLL	Obinutuzumab BTKis or venetoclax ± rituximab
	WM/MZL	Rituximab ± chemotherapy BTKis or venetoclax
Paraproteinemic neuropathy (anti-MAG antibody negative) Cryoglobulinemia	IgM MGUS	Rituximab
	CLL/WM/MZL	Rituximab ± chemotherapy ± bortezomib BTKis or venetoclax
POEMS syndrome AL amyloidosis	IgG or IgA MGUS	Bortezomib Lenalidomide
	IgG or IgA MM	Daratumumab ± bortezomib ± IMIDs Autologous stem cell transplant
Castleman’s diseases		Siltuximab
Intravascular lymphoma neurolymphomatosis		Rituximab-based chemotherapy

CLL—chronic lymphocytic leukemia; WM—Waldenstron’s macroglobulinemia; MZL—marginal zone lymphoma; MGUS—monoclonal gammopathies of undetermined significance; MAG—myelin-associated-glycoprotein; POEMS— Polyneuropathy, Organomegaly, Endocrinopathy, Monoclonal gammopathy, Skin changes; BTK—Bruton’s tyrosine kinase.

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
