# Peer review of "From Biology to Treatment of Monoclonal Gammopathies of Neurological Significance"

_cancers, 2022, doi:10.3390/cancers14061562_

Round 1

Reviewer 1 Report

The authors of this paper review the most relevant patterns of monoclonal gammopathy of neurological significance, with a focus on the known or supposed pathogenic mechanisms that could be targeted by novel therapies.

I think it would be very useful for the reader:

  • to include an explanation of the diagnostic flowcharts depicted in figures number 2 and 3.
  • to include an schema summarizing the treatment of all the entities included in the review.

Other minor points:

INTRODUCTION

  • Line 4: during the course of either pre-neoplastic diseaseS … or NEOPLASTIC hematological diseases…
  • Line 8: syndrome is repeated. Please, move the meaning of POEMS (between brackets) after the word syndrome.

SIGNALING PATHWAYS IN B-CELL MALIGNANCIES

  • Please rephrase since “Ibrutinib and other BTK inhibitors have NOT became the standard of care of MOST B-cell malignancies”.
  • Consider to explain the mechanism of action of venetoclax to better understand the parapraph.
  • At the end of the paragraph: On the other hand, a bone marrow BIOPSY is recommended…

MONOCLONAL GAMMOPATHIES OF NEUROLOGICAL SIGNIFICANCE

  • Consider the abbreviation of peripheral neuropathy as PN
  • A multidisciplinary assessment involving hematologistS, neurologistS, radiologistS,…

Monoclonal gammopathies of undetermined significance

  • Please, complete the definition of MGUS in the first paragraph.
  • Please rephrase the sentence “The most common MGUS subtype is IgG MGUS but it is associated with a low risk to develop MM” since IgG is the isotype with the lower risk of progression but there are also other prognostic factors such as the amount of IgG as well as the FLC ratio that could worsen the prognosis of an otherwise IgG MGUS patient.
  • Third paragraph: First line: has favored

WM/LPL

  • …More rarely IgA or IgG
  • Cryoglobulinemias can be classified as followS
  • Please check this: IgM>IgG>IgG

Anti-MAG antibody neuropathy

  • These new drugs selectively target

AL amyloidosis

  • Second paragraph: please check “polyneuropathy with axonal polyneuropathy”
  • I would not consider IMIDs as amyloid-targeted treatments and in fact I would say that all of them together with proteasome inhibitors and even MoAbs have merely been “adopted” from the MM field. Also, it is important to highlight that thalidomide y bortezomib are actually associated with PN and therefore should be avoided if possible as a treatment of patients with PN.

POEMS

  • Please check “whithe nails”
  • Last paragraph: Secondo; To date Three

Castleman´s disease

  • Most of the published data derive

NHL-RELATED PERIPHERAL NEUROPATHIES

CLL

  • Mainly affectS
  • Line 21: Showing persistent hematological and neurological ??

Intravascular lymphoma

  • Line 5: ConstitutiveLY

CONCLUSIONS

  • ManifestationS
  • Makes (it)
  • Increasing evidence supportS
  • TargetED therapies

Reviewer 2 Report

The review by Visentin et al. on monoclonal gammopathies of neurological significance covers pathophysiology, diagnosis, patient management and treatments. It is very clear, well written, interesting and informative. The figures are clear and appropriate, the bibliography is quite complete.

This reviewer can only recommend publication. 

Author Response

We are honored to known that reviewer #2 appreciates our manuscript.

Reviewer 3 Report

This paper aims to summarise the spectrum of MGNS, their distinctive clinical and neurophysiological phenotypes, pathophysiological events and new therapeutic approaches.

The authors do well to cover an extensive topic, but this also serves to undermine the paper. 

The summary of the biological aspects is nicely addressed, bringing out the key molecular underpinnings of these conditions. However, it comes across as 2 parallel topics rather than making more of the molecular mechanisms in the pathophysiology of the various entities.

Attempting to cover the biology, pathophysiology, clinical characteristics and treatment is perhaps too ambitious.

I would recommend a tighter focus, removal of some repetition in different sections and zeroing in on the key connections between the biology -pathophysiology- treatment.

The authors could reference other papers for the clinical descriptions of the various entities and focus on the translational aspects of the conditions.

Author Response

  • The authors do well to cover an extensive topic, but this also serves to undermine the paper. The summary of the biological aspects is nicely addressed, bringing out the key molecular underpinnings of these conditions. However, it comes across as 2 parallel topics rather than making more of the molecular mechanisms in the pathophysiology of the various entities Attempting to cover the biology, pathophysiology, clinical characteristics and treatment is perhaps too ambitious. I would recommend a tighter focus, removal of some repetition in different sections and zeroing in on the key connections between the biology -pathophysiology- treatment. The authors could reference other papers for the clinical descriptions of the various entities and focus on the translational aspects of the conditions

We thank the reviewer for his punctual and kind observations. Following his/her suggestions, we removed repetitions and add some references to the clinical pictures of the diseases. However, we have still maintained some descriptions since we believe it would be useful for readers and the lack some peculiar signs and symptoms might jeopardize the review.

Round 2

Reviewer 3 Report

The revisions have been helfpul